# Adhesive, Transparent Tannic Acid@ Sulfonated Lignin-PAM Ionic Conductive Hydrogel Electrode with Anti-UV, Antibacterial and Mild Antioxidant Function

**DOI:** 10.3390/ma12244135

**Published:** 2019-12-10

**Authors:** Qinhua Wang, Hui Zhang, Xiaofeng Pan, Xiaojuan Ma, Shilin Cao, Yonghao Ni

**Affiliations:** College of Material Engineering, Fujian Agriculture and Forestry University, Fuzhou 350002, China; fafuqhw@163.com (Q.W.); 17862266728@163.com (H.Z.); 1171092015@fafu.edu.cn (X.P.)

**Keywords:** sulfonated lignin, tannic acid, electronic materials, functional, PAM

## Abstract

Inspired by mussel adhesion chemistry and ion electronics, a novel Ca^2+^-tannic acid@ sulfonated lignin-polyacrylamide (TA@SL-PAM) hydrogel was prepared via Ca^2+^-TA@SL composites and the PAM system, where a Ca^2+^-TA@SL composite was fabricated via TA doping with SL and the subsequent adsorption of Ca^2+^. The properties of the hydrogel were thoroughly investigated and the hydrogel was presented as multifunctional. The introduction of Ca^2+^-TA@SL composites endowed the hydrogel with excellent conductivity, adhesion and ultraviolet (UV) resistance, and improved antioxidant and antibacterial properties. More importantly, the Ca^2+^-TA@SL-PAM hydrogel electrode could accurately detect physiological signals of human (e.g., electrocardiogram (ECG), electromyography (EMG).

## 1. Introduction

Recently, conductive hydrogels (CHs) have captured the interest of scholars in the field of wearable electronics for emerging human–machine interfaces [1]. CHs integrated with excellent electrical conductivity, skin compatibility, and biocompatibility have become potential candidates for flexible medical electronics [2]; for example, flexible electrodes can detect physiological signals in humans. For the construction of visual medical products, it is urgently required that the CHs have great conductivity, adhesion, and transparency [3]. Moreover, the antibacterial and antioxidant properties are advantageous to CHs in preparing static electrodes suitable for long-term physiological monitoring. More importantly, ultraviolet (UV) protection seems to be essential because people are extremely susceptible to UV radiation from sunlight [4]. Unfortunately, it is difficult to prepare a hydrogel combing multi-functional properties simultaneously.

Here, inspired by mussel adhesion chemistry and ion electronics, we tried to use tannic acid (TA), a dopamine-like substance, to be doped with sulfonated lignin (SL) to construct composites base with adhesive properties. To further enhance the conductivity, Ca^2+^ (CaCl_2_) was introduced into the composite through chelation. As far as we know, TA with a large amount of catechol structures can adsorb biocompatible Ca^2+^.

Subsequently, the above Ca^2+^-TA@SL composites were incorporated into the polyacrylamide (PAM) hydrogel network, which can endow the hydrogel with great adhesion and conductivity. Meanwhile, TA and SL can impart the hydrogel with anti-UV and mild anti-bacterial ability. In addition, the TA can enhance the antioxidant properties of hydrogel. Importantly, the Ca^2+^-TA@SL-PAM CHs, as a non-invasive adhesive electrode, can achieve the accurate capture of human ECG and EMG signals. We believe that biomass-based CHs can provide more application ideas in the field of human-machine interfaces.

## 2. Experimental

Lignin was separated from the black liquor of radiated pine Kraft pulping; SL was prepared via lignin sulfonation [5]. TA@SL composite was prepared via tannic acid (TA) doping with sulfonated lignin (SL); thereafter, Ca^2+^ was absorbed onto the TA@SL composite. Next, acrylamide (AM), ammonium persulfate (APS), N, N’-Methylene bisacrylamide (BIS), and tetramethylethylenediamine (TMEDA) were added to the Ca^2+^-TA@SL suspension to form Ca^2+^-TA@SL-PAM hydrogel. The detailed preparation procedure and composition of the Ca^2+^-TA@SL-PAM hydrogels were listed in Appendix A (Appendix A).

The morphology of the SL, TA@SL and Ca^2+^-TA@SL-PAM hydrogel were observed by field-emission scanning electron microscopy (Nova Nano SEM 230, Brno, Czech Republic), the element distribution of TA@SL was observed by Octane Elect EDS (EDAX, Mahwah, NJ, USA). The visible (550 nm) and UV (365 nm) transmittance of the PAM and Ca^2+^-TA@SL-PAM hydrogels with 3 mm thicknesses were measured using an optical transmittance meter (LS108H, Shenzhen, China). The hydrogel conductivity was measured by an LCR meter. The conductivity (*σ*) is calculated by *σ* = d/*RS*, in which *R* is the resistance, and d and *S* are the hydrogel’ geometry factors corresponding to the length and cross-sectional area, respectively. Detailed antibacterial and antioxidant assay of Ca^2+^-TA@SL-PAM hydrogel can be found in the Appendix A. The adhesive strength (fresh pigskin) of PAM-5 hydrogel was tested via the lap-shear testing method [5].

The PAM-5 hydrogel was cut into cubes and then adhered to specific skin locations. The ECG (electrocardiograph) and EMG (electromyography) signals of volunteer were collected using a multi-channel bio-signal acquisition and process system (RM6240CD, Chengdu, China).

## 3. Results and Discussion

Figure 1a displays the complete synthetic route for the Ca^2+^-TA@SL-PAM hydrogel. Figure 1b shows the rice-like structure of SL. TA was doped with SL under alkaline conditions and the TA@SL composite was obtained (Figure 1c). Elemental analysis indicated that the characteristic S elements from the SL sulfonate groups were uniformly distributed on the TA@SL composite. Figure 1d demonstrates that the synthesized TA@SL particles contain more negatively charged groups than pure SL and TA particles. Therefore, the changes in the zeta potential of the various particles also confirmed the successful functionalization and assembly of the TA@SL composite. Then, Ca^2+^ was successfully absorbed onto the TA@SL composite through complexation with catechol groups on TA@SL composites [6]. Finally, AM, APS, MBA, and TMEDA were added to the system for in situ polymerization to form the Ca^2+^-TA@SL-PAM hydrogel. Figure 1e and elemental analysis (Figure 1f) present that TA@SL particles are evenly distributed on the surface of the nanocomposite hydrogel, which may provide a structural basis for the subsequent functionalization of the hydrogel.

Figure 2a presents the synergy effects of the different components on the Ca^2+^-TA@SL-PAM hydrogel, making the resultant hydrogel have enhanced conductivity, adhesion, transparency, antioxidant, antibacterial, and anti-UV properties. As shown in Figure 2b, the raw PAM hydrogel is transparent and the transmittance was above 92%. Unfortunately, it has a transmittance of 91% at 365 nm and is substantially incapable of filtering UV radiation. With an increase of TA content on the TA@SL composite, the UV transmission rate of TA@SL-PAM hydrogel decreased from 20% to nearly 0%. Meanwhile, the transmittance (550 nm) of the hydrogel remained above 72%. For instance, the palm prints are clearly seen through the PAM-5 hydrogel. The anti-UV effect of the hydrogel is mostly derived from the absorption of UV light by the benzene ring (from SL and TA) [4]. Figure 2c shows that the TA@SL-PAM hydrogel can use conductive Ca^2+^ to achieve the circuit connectivity (0.91 S/m). The interaction of catechol groups on the TA and Ca^2+^ inspired by mussel adhesion chemistry can impart adhesion to the hydrogel [7]. As presented in Figure 2d, PAM-5 hydrogels can be bonded to different materials such as skin, foam, plastic, iron, rubber, and glass. As revealed in Figure 2e, the PAM hydrogel shows no effect on scavenging staphylococcus aureus (*S. aureus*), while the SL based PAM hydrogel and Ca^2+^-TA@SL-PAM hydrogel can scavenge some of the *S. aureus*. Notably, PAM-5 hydrogel can even achieve a 98% *S. aureus* killing rate. Figure 2f indicates the UV-Vis absorption curve of the 1,1-diphenyl-2-picrylhydrazyl (DPPH) free radical solution in the control group and Ca^2+^-TA@SL-PAM hydrogel with different TA@SL content. Due to the presence of odd electrons, the DPPH radical generates a strong absorption peak at 517 nm [8]. However, the coordination of electrons with the hydrogen atoms in the antioxidant results in a decrease in the intensity of the absorption peak. It is clearly seen that the absorption intensity decreases with increasing TA@SL content in Ca^2+^-TA@SL-PAM hydrogel. Compared with the control group, the DPPH scavenging rate even reached 75% when the TA@SL content reached 16 mg/mL (Figure 2g). These results demonstrate that the Ca^2+^-TA@SL-PAM hydrogel has effective antioxidant properties. Generally, the antioxidant ability of the hydrogel is derived from the action of TA and lignin; the phenolic hydroxyl group on TA and SL can reduce the free DPPH radical to diphenyl-picrylhydrazine [9].

Biocompatible Ca^2+^ have a greater advantage in the preparation of non-invasive electronics [10]. Herein, the suitability of ion-conductive Ca^2+^-TA@SL-PAM hydrogel electrodes was evaluated in detail. As presented in Figure 3a, the hydrogel could be firmly adhered on the skin and the even arm shook vigorously (the adhesion strength between hydrogel and fresh pig skin is 4.31 KPa). Moreover, the adhesive hydrogel could be completely peeled off and had no residue on the arm. Importantly, the adhesive electrodes attached to the palm muscle of the female volunteer’s arm achieved accurate collection of epidermal electrical signal changes caused by the volunteers’ arm-clamping, relaxation, and straightening muscle switching (Figure 3b). Similarly, volunteers’ complete ECG and heart rate could also be accurately monitored by the adhesive electrodes (Figure 3c) [5]. Adhesive hydrogel can increase the service life of the electrode and prevent it from falling off the skin, thereby affecting the stability of the signal.

## 4. Conclusions

A novel PAM based ionic conductive hydrogel electrode was successfully developed—inspired by mussel and ion electronics. The TA@SL composite application endowed the hydrogel with excellent adhesion, UV resistance, antioxidant, and moderate antibacterial properties, but without the destruction of transparency. The Ca^2+^ introduction imparted the hydrogel with electrical conductivity. The fabricated hydrogel electrode could accurately collect the physiological signals from the human, including EMG and ECG. Considering the prominent properties and quick feedback of the human motion, the Ca^2+^-TA@SL-PAM hydrogel electrode has great potential in visual medical electronics.

## Figures and Tables

**Figure 1 materials-12-04135-f001:**
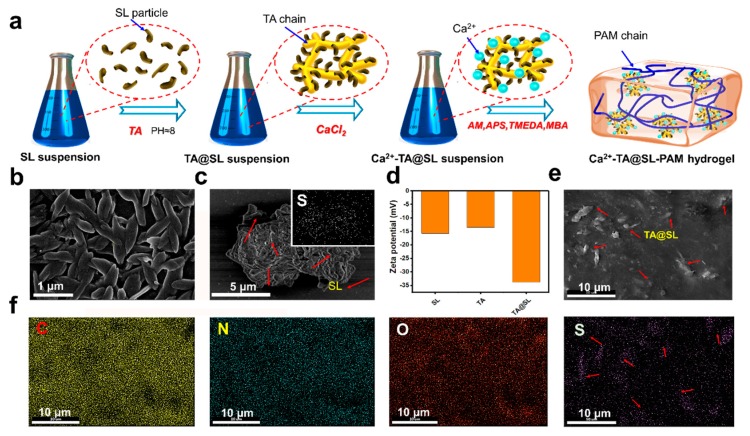
(**a**) Synthesis route of Ca^2+^-TA @SL-PAM hydrogel. SEM of (**b**) SL particles and (**c**) TA@SL composites and distribution of S elements on the TA@SL composite. (**d**) Zeta potentials of SL, TA, and TA@SL particles. (**e**) Distribution of TA@SL composites on the surface of Ca^2+^-TA@SL-PAM hydrogel. (**f**) Elemental distribution mapping of the Ca^2+^-TA@SL-PAM hydrogel surface.

**Figure 2 materials-12-04135-f002:**
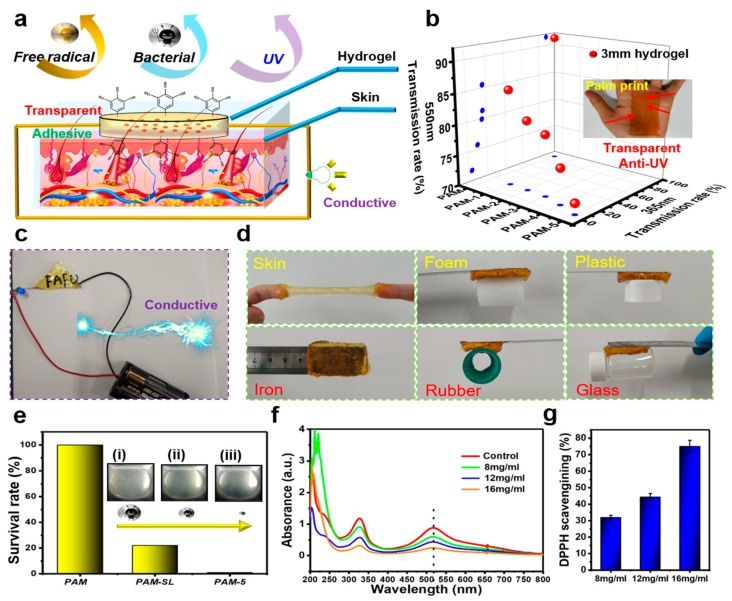
(**a**) Cartoon illustration of the multifunctional Ca^2+^-TA@SL-PAM hydrogel. (**b**) The transmittance of Ca^2+^-TA@SL-PAM hydrogel (**c**) PAM-5 can be used as a circuit wire. (**d**) PAM-5 can be attached to different materials. (**e**) Survival rate of *S. aureus* grown on different hydrogels. (**f**) Absorbance change of DPPH when exposed to different hydrogels. (**g**) DPPH scavenging rate.

**Figure 3 materials-12-04135-f003:**
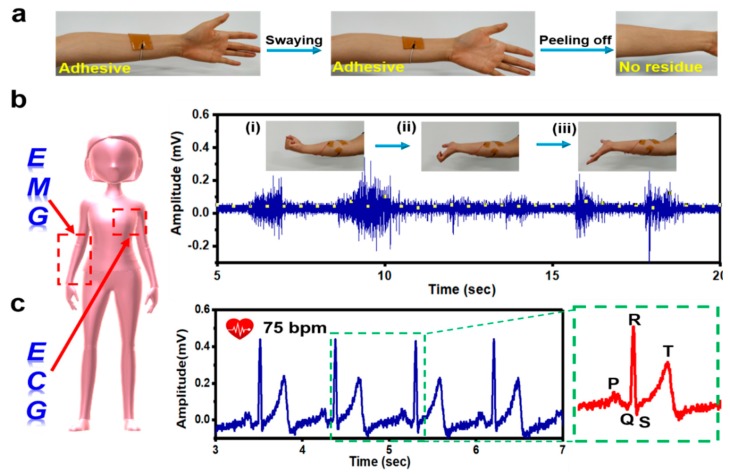
(**a**) PAM-5 adheres tightly to the author’s skin and shakes sharply without falling off. (**b**) The electrode collects variations in EMG signals. (**c**) The electrode can monitor ECG signals of human.

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
