# Peer review of "Adhesive, Transparent Tannic Acid@ Sulfonated Lignin-PAM Ionic Conductive Hydrogel Electrode with Anti-UV, Antibacterial and Mild Antioxidant Function"

_materials, 2019, doi:10.3390/ma12244135_

Round 1

Reviewer 1 Report

The paper can be accepted for the publication in Materials after a major revision concerning the following points:

1) The obtained materials should be better characterized. The Ca2+TA@SL-PAM hydrogel in particular shoul be better investigated and chracterized.

2) There are both in the letter than in the supplementary material several not-explained acronyms. The authors should provide the meaning.

Author Response

Reviewers' Comments to Author:
Reviewer: 1

The paper can be accepted for the publication in Materials after a major revision concerning the following points:

1) The obtained materials should be better characterized. The Ca2+-TA@SL-PAM hydrogel should be better investigated and characterized.

Author’s response:

Thanks for the valuable suggestions. As you suggested, we have added some other experimental characterization and discussion. First, the zeta potentials of sulfonated lignin (SL) and tannic acid (TA)@SL particles were measured, which may further demonstrate the physical differences between the hydrogels before and after the in-situ coating reaction. In addition to the various functions of the Ca2+-TA@SL- polyacrylamide (PAM) hydrogel described above; we also explored the distribution of TA@SL particles in the hydrogel system (SEM image) to provide structural support for the construction of functionalized composite hydrogels. Finally, the conductivity of the composite hydrogel is also measured, which may be useful for quantifying the conductivity of the hydrogel.

Figure 1d demonstrates that the synthesized TA@SL particles contain more negatively charged groups than pure SL and TA particles. Therefore, the changes in the zeta potential of the various particles confirmed that functionalization and assembly were successful.

Figure 1e and elemental analysis present that TA @ SL particles are evenly distributed on the surface of the nanocomposite hydrogel, which may provide a structural basis for the subsequent functionalization of the hydrogel.

2) There are both in the letter than in the supplementary material several not-explained acronyms. The authors should provide the meaning.

Author’s response:

Thanks for your comment.

Your suggestion is very necessary to improve the readability of the article. In order to improve the rigor of the manuscript, we explained the first occurrence of the acronyms in the abstract, main text and supplementary material.

Reviewer 2 Report

Summary:

The manuscript presents the development of a composite hydrogel with several potentially favorable properties including electrical conductivity, adhesion, biocompatibility, and antibacterial properties. Overall, has a reasonable amount of data for a report and seems to achieve the properties that it set out as advantageous. Whether these truly reflect clinical needs or not, remains up for debate. After some revisions, I believe that this manuscript would be suitable for publication. Please see specific comments below.

Major issues:

The authors have included human data from a volunteer. While I have no reason to suspect that this experiment was performed on a non-consenting participant, it is important that the authors include the information about the permission they received from an Institutional Review Board to ensure that ethical standards were met. I do not understand why UV absorbance, but visible light transmission is important. Lines 28-29 state that patients in the ICU are exposed to substantial UV radiation, but I do not believe that to be true and it is not clear from the citation. Perhaps this could be cleared up with additional explanation and references. Also, it seems that the primary application for this gel would be in wearables that would not necessarily be limited to in-hospital use. Calling this material antibacterial seems like a substantial overstatement. Bacteria does not grow on the PAM-5 as well as the PAM, but that does not make it antibacterial. It is important to note that there are still colonies formed and that antibacterial is a highly relative term in this case. It would be good to have a true negative control with bacteria cultured on a surface that does not allow for their survival. Similarly, the y-axis label on Figure 2E is not accurate. It should really be something like “Percent of colonies compared to PAM.” It would be useful to address how motion affects the ECG and EMG signals (beyond the response to the muscle that is being exercised for EMG). Would this confound the signal measured? I suspect so. Please discuss the potential limitations of this approach or how these issues are mitigated.

Minor issues:

The authors really need to define their acronyms the first time they are used. At present, the abstract is unreadable for this reason. Ca and UV are fine, but the rest should be defined. For example, “PAM” is never defined, so I will assume it means polyacrylamide and not protospacer adjacent motif. AM, APS, MBA, TMEDA have the same issue. Alternatively, S. aureus is defined twice. Grammar needs to be improved prior to publication. Figure 1D: What scale was this image taken at? The same as magnified 1C? Please include a scale bar.

Author Response

Reviewer: 2

Summary:

The manuscript presents the development of a composite hydrogel with several potentially favorable properties including electrical conductivity, adhesion, biocompatibility, and antibacterial properties. Overall, has a reasonable amount of data for a report and seems to achieve the properties that it set out as advantageous. Whether these truly reflect clinical needs or not, remains up for debate. After some revisions, I believe that this manuscript would be suitable for publication. Please see specific comments below.

Major issues:

1)The authors have included human data from a volunteer. While I have no reason to suspect that this experiment was performed on a non-consenting participant, it is important that the authors include the information about the permission they received from an Institutional Review Board to ensure that ethical standards were met.

Author’s response:

Thanks for your comment.

Respect for human rights is very important. However, this experiment is different from clinical drugs and human pathology experiments. It is non-invasive, does not swallow, and only touches the skin of volunteers' arms. Similarly, many wearable sensor and electrode studies do not provide an ethical license and may therefore differ from medical human experiments (Chem. Mater. 2019, 31, 15, 5625-5632Science Advances  15 Jun 2018:Vol. 4, no. 6, eaat0098) . In addition, we also consulted the hospital’s ethics department (900 Hospital of the Joint Logistics Team, Fuzhou, China), who believe that our experiments do not have or do not require ethical qualifications. Therefore, we express our regret and will pay attention to this issue in future experiments. But here, we affirm that volunteers are one of the authors and are willing to contact these materials. As of today, she has no skin discomfort. Thank you very much for your kind reminder.